# The Real-Time Estimation of Respiratory Flow and Mask Leakage in a PAPR Using a Single Differential-Pressure Sensor and Microcontroller-Based Smartphone Interface in the Development of a Public-Oriented Powered Air-Purifying Respirator as an Alternative to Lockdown Measures

**DOI:** 10.3390/s25175340

**Published:** 2025-08-28

**Authors:** Yusaku Fujii

**Affiliations:** School of Science and Technology, Gunma University, Kiryu 376-8515, Japan; fujii@gunma-u.ac.jp; Tel.: +81-277-30-1756

**Keywords:** powered air-purifying respirator (PAPR), PAPR for everyone, alternative to lockdown, respiratory flow estimation, mask leakage detection, differential pressure sensor, Arduino microcontroller, Bluetooth Low Energy (BLE), smartphone interface, pandemic preparedness, airborne infection control

## Abstract

**Highlights:**

**What are the main findings?**
A low-cost PAPR system was developed for general public use, featuring the real-time estimation of respiratory flow (***Q*_3e_**) and mask leakage using only a single differential-pressure sensor and battery voltage.The estimated respiratory flow closely matched the reference sensor measurement (R^2^ = 0.952) and the system successfully detected improper wearing based on the time-integrated ***Q*_3e_**.

**What is the implication of the main finding?**
This simple sensing framework enables the practical and scalable visualization of wearing and leak status in public-oriented PAPRs, contributing to airborne infection control without lockdowns.The platform can be expanded to support pressure-assisted breathing, biosensing, and societal-scale infection risk monitoring via a PAPR Wearing Status Network Management System.

**Abstract:**

In this study, a prototype system was developed as a potential alternative to lockdown measures against the spread of airborne infectious diseases such as COVID-19. The system integrates real-time estimation functions for respiratory flow and mask leakage into a low-cost powered air-purifying respirator (PAPR) designed for the general public. Using only a single differential-pressure sensor (SDP810) and a controller (Arduino UNO R4 WiFi), the respiratory flow (***Q*_3e_**) is estimated from the differential pressure (**Δ*P***) and battery voltage (**V_b_**), and both the wearing status and leak status are transmitted to and displayed on a smartphone application. For evaluation, a testbench called the Respiratory Airflow Testbench was constructed by connecting a cylinder–piston drive to a mannequin head to simulate realistic wearing conditions. The estimated respiratory flow ***Q*_3e_**, calculated solely from **Δ*P*** and **V_b_**, showed high agreement with the measured flow ***Q*_3m_** obtained from a reference flow sensor, confirming the effectiveness of the estimation algorithm. Furthermore, an automatic leak detection method based on the time-integrated value of ***Q*_3e_** was implemented, enabling the detection of improper wearing. This system thus achieves respiratory flow estimation and leakage detection based only on **Δ*P*** and **V_b_**. In the future, it is expected to be extended to applications such as pressure control synchronized with breathing activity and health monitoring based on respiratory and coughing analysis. This platform also has the potential to serve as the foundation of a PAPR Wearing Status Network Management System, which will contribute to societal-level infection control through the networked sharing of wearing status information.

## 1. Introduction

In this study, a prototype system of a powered air-purifying respirator (PAPR), which was originally designed as a low-cost device for the general public as an alternative to lockdown measures [1,2], was developed by modifying it to include estimation functions for respiratory flow and mask leakage, using a single differential-pressure sensor and a controller (microcontroller board: Arduino UNO R4 WiFi). The wearing status and leak status are displayed in real time on a smartphone application via Bluetooth Low Energy (BLE) communication.

Evaluation using a cylinder–piston-driven mannequin head confirmed that the disturbance flow (***Q*_3e_**), estimated solely from the differential pressure **Δ*P*** and battery voltage **V_b_**, showed high agreement with the measured value (***Q*_3m_**). In addition, a method for the automatic judgment of wearing and leakage based on the time-integrated value of ***Q*_3e_** was implemented.

In the future, this system is expected to be applied to societal-level infection control, particularly in extensions such as pressure control to assist breathing activity and health monitoring based on cough and respiratory pattern analysis.

The COVID-19 pandemic that began in early 2020 had a serious impact on societies worldwide [3,4]. Even in the future, the emergence of variants or novel airborne infectious diseases may cause new pandemics [5]. While vaccines and therapeutic agents are available, non-pharmaceutical public health measures (PHSMs) such as physical distancing, mask-wearing, ventilation, event restrictions, travel restrictions, and quarantine have been widely implemented. However, if response measures are delayed, lockdowns may be enforced [6,7], which impose severe social and economic burdens. Therefore, the establishment of a strong and sustainable infection control method as an alternative to lockdown is an urgent challenge.

Among transmission routes, in addition to contact, oral, and droplet infection, airborne transmission exists. Airborne transmission via aerosols is especially difficult to contain as aerosols can float and diffuse widely in indoor spaces [8,9]. It is highly likely that airborne transmission will be a major route in the next pandemic. In particular, the dynamics of infection via aerosol particles cannot be accurately represented by conventional SIR models, and more sophisticated modeling is required [10]. However, except for vaccines, there are few effective countermeasures against airborne infection that are accessible to the general public, and even these have limited efficacy [11].

Medical-grade powered air-purifying respirators (PAPRs) are widely used as protective equipment with high performance against airborne infectious diseases. First, PAPRs are equipped with high-performance nonwoven filters such as HEPA filters. According to the United States Environmental Protection Agency (EPA), HEPA filters guarantee a collection efficiency of 99.97% or higher for the most penetrating particle size (MPPS), typically 0.3 µm [12]. This means they also perform well in removing both larger and smaller particles than those 0.3 µm in size.

Recent studies have shown that the MPPS may vary depending on factors such as the filter material, electrostatic charge, and airflow rate; in some cases, MPPS can be as small as 0.1–0.2 µm for non-electret filters, or even around 40 nm for electret types [13]. While the MPPS cannot be strictly defined across all scenarios, HEPA filters have long been verified under rigorous standards and consistently demonstrate high performance in practical settings [12]. Therefore, for high-reliability applications such as PAPR, HEPA filters remain an extremely effective protective measure.

Second, PAPRs maintain a positive-pressure environment inside the hood, which effectively prevents the intrusion of external aerosols. This pressure-based protection is especially prominent in helmet/hood-type PAPRs. According to manufacturer certification based on WPF or SWPF testing, Assigned Protection Factors (APFs) can reach up to 1000 [14,15], which is orders of magnitude higher than those for typical half-mask respirators (APF = 10) or N95 masks (equivalent to APF = 10). In contrast, negative-pressure respirators such as N95 masks may allow inward airflow through face gaps due to the inward negative pressure generated during inhalation. Note that APF is defined by OSHA as the ratio of external to internal concentration: APF = [external concentration]/[internal concentration] [14].

Third, the use of PAPRs is expected to contribute to reducing infection risk by significantly lowering the number of virus particles inhaled. Although the infectious dose required to cause infection varies depending on the pathogen type, host immunity, and exposure route, it has inherent uncertainty [16]. Nevertheless, for COVID-19, the analysis of actual superspreader events has estimated that the infectious dose (N_0_) is roughly between 300 and 2000 virus particles [17]. Under such conditions, the use of PAPRs combining HEPA filtration and the positive-pressure suppression of external air inflow is highly likely to reduce inhaled particles below the infection threshold.

Therefore, from the standpoint of reducing the probability of infection establishment, PAPR use is expected to significantly contribute to infection risk reduction. The authors have prototyped low-cost, simplified PAPRs while maintaining performance equivalent to medical-grade devices and have examined the technical and social issues toward societal implementation. This initiative has three main contributions:(1)It diversifies infection control tools by introducing a new countermeasure alongside vaccines and behavioral changes.(2)It offers a practical alternative to lockdown, reducing social and economic losses.(3)It provides a sustainable infrastructure for future pandemic preparedness through the design and deployment of low-cost PAPRs for everyday use.

Thus, this paper proposes a new option for airborne infection control and contributes to enhancing societal resilience.

The superior performance of PAPRs compared to standard face masks lies in the positive pressure maintained inside the hood, which physically blocks inward airflow through facial gaps. Moreover, PAPRs remove virus-laden particles through mechanical filtration, allowing virus-independent protection.

The authors’ prototype PAPR [18] is equipped with a HEPA filter, fan, and battery, with a total component cost of about 40 USD. The system supplies purified air via a HEPA filter (removal efficiency ≥99.97% at 0.3 μm) into the hood to maintain positive pressure. Even if slight gaps form between the hood and the face, the direct inflow of unfiltered air is prevented. Exhaust air passes through nonwoven filters via natural pressure-driven flow, and even if the user is infected, this design is expected to suppress viral emission. This basic functionality is equivalent to that of commercial medical-grade PAPRs.

Although PAPRs provide high protective performance, it is important to clarify their societal significance by comparing them with other common protective equipment. The prototype PAPR developed in this study has unique value not only in terms of performance but also in cost and usability compared with existing N95 masks and medical-grade PAPRs. N95 masks are inexpensive and widely used, but they face challenges in terms of fit and comfort during long-term wear. On the other hand, commercial medical-grade PAPRs provide high protection but are priced on the order of 1000 USD, and their weight and wearing configuration impose limitations on daily use by the general public.

The prototype PAPR presented in this study maintains protective performance equivalent to that of medical-grade PAPRs while ensuring comfort through weight reduction and structural simplification and achieving low cost by utilizing commercially available components. In addition, the positive-pressure design reduces breathing resistance, thereby balancing user safety with reduced physiological burden. As a result, this system opens the possibility of a sustainable protective device suitable not only for medical settings but also for industrial environments and daily use by the general public.

This study was part of a project to develop a public-oriented PAPR as a socially deployable technology to replace lockdowns in the control of airborne infectious diseases such as COVID-19. In particular, this paper focuses on the development and validation of a real-time estimation method for respiratory flow and wearing conditions aimed at improving wearing comfort—an important issue in PAPR practical use.

In the authors’ previous study [19], the airflow responses of the supply unit and exhaust filter were analyzed by connecting them to separate pressure buffers and evaluating their fluid response using a respiratory simulator. In contrast, this study constructed a new test platform—the “Respiratory Airflow Testbench”—in which the PAPR was worn directly on a mannequin head equipped with a respiratory simulator to more realistically evaluate the actual wearing state.

The prototype PAPR includes an Arduino UNO R4 WiFi controller and a differential-pressure sensor (SDP810), enabling the real-time estimation of respiratory flow ***Q*_3e_** based on **Δ*P*** and **V_b_**. The accuracy of ***Q*_3e_** was verified by comparing it with measured values ***Q*_3m_** obtained from a flow sensor. Furthermore, a BLE-based smartphone application was developed that displays the wearing status and leak status in real time, providing feedback to the user.

These features suggest the feasibility of extensions such as dynamic pressure control based on breathing activity and the detection of abnormal breathing or coughs. In the future, a PAPR Wearing Status Network Management System [1,2] is envisioned to visualize and share risk at the societal level through the networking of wearing data.

## 2. Development of a PAPR Prototype with Respiratory Flow Estimation and Leakage Detection Functions

Figure 1 shows the prototype of a powered air-purifying respirator (PAPR) equipped with respiratory flow estimation and leakage detection functions. A lightweight work helmet shell was partially cut and modified to serve as a support structure for the hood and the intake/exhaust units.

An “air-supply unit (airtight room)” was constructed using white styrene board with a thickness of 5 mm, and a Sirocco fan (model: San Ace B76, manufacturer: Sanyo Denki Corp., Tokyo, Japan, maximum static pressure: 150 Pa, maximum flow rate: 360 L/min) was installed inside.

A transparent PVC sheet (thickness: 0.5 mm) was used for the shield (transparent window), and a Tyvek airtight cap was cut and used for the hood part. The tension of the elastic sealing cord was adjusted to prevent any gaps between the hood and the user’s face.

At the front of the air-supply unit, a HEPA filter for air purifiers (PMMS-DCHF, Iris Oyama Ltd., Sendai, Japan) with an effective surface area of approximately 700 cm^2^ was installed. A thin nonwoven fabric filter (approx. 25 cm^2^ surface area) was positioned in front of the ears as the exhaust filter in a configuration that allowed the ears to remain exposed for acoustic considerations.

General-purpose adhesives and hot-melt adhesives (HMAs) were used to bond the shell components such as the supply unit, exhaust filter, shield, and hood.

The controller used was an Arduino UNO R4 WiFi. The differential pressure between the inside of the hood and the external air was measured using a differential-pressure sensor (Model: SDP810-500 Pa, manufacturer: Sensirion, Stäfa, Switzerland). This sensor was connected via a Qwiic connector through a multiplexer (model: 8ch I2C Hub TCA9548A, manufacturer: Grove, San Francisco, CA, USA). The multiplexer allowed additional reference flowmeters to be connected later.

The battery voltage was stepped down using four resistors (2.5 kΩ each) connected in series and measured via the analog input pin of the controller. Power for both the fan and the controller was supplied by a lithium-ion battery (rated 12 V) mounted on the back of the helmet shell.

In terms of airflow, purified air is supplied into the hood by the supply unit, maintaining positive pressure within the hood. Exhaust air is discharged through the exhaust filter due to the pressure difference.

The controller transmits the judged wearing status (ON/OFF) and leakage status (Leak/No Leak) to a dedicated Android smartphone application (“PAPR Wearing Status Monitor”) at regular intervals (default: approximately every 3 s) via Bluetooth Low Energy (BLE). As explained later, wearing status is determined using only the measured values of differential pressure **Δ*P*** [Pa] and battery voltage **V_b_** [V].

The prototype PAPR was developed by the author as a custom system assembled from commercially available components. Its total mass was approximately 656.1 g, including the battery (174.4 g), and the continuous operating time was about 320 min (5 h 20 min). The operating time can be extended depending on the battery capacity, and even for this prototype, more than 6 h of use is expected with capacity expansion.

For comparison, the commercially available PAPR “Versaflo TR-300+” manufactured by 3M (Maplewood, MN, USA, listed in [19], Table 1 [A]) has a total mass of approximately 1419.6 g, consisting of the belt-mounted air-supply unit (about 1107.7 g), the supply tube (about 201.9 g), and the hood (about 110.0 g). The fact that the mass of the prototype PAPR is less than half of this value is attributed to its integrated helmet-type structure and the use of lightweight materials, which contribute to improved comfort during long-term wear. However, for product development, it will be necessary to ensure an optimal balance between strength and weight.

The smartphone application “PAPR Wearing Status Monitor” receives and displays the status transmitted by the controller. When the wearing status is OFF, the screen background is white. When wearing is ON and leakage is detected (YES), the screen turns red (warning). When leakage is not detected (NO), the screen turns blue (normal) (see Figure 1).

Figure 2 shows the results of measurements of the airflow rate ***Q*_1_** [L/min] of the supply unit as a function of differential pressure **Δ*P*** [Pa] and battery voltage **V_b_** [V], based on the methods and apparatus described in [20]. Measurements were performed at battery voltages **V_b_** = 11.00, 12.00, and 13.00 V. The following polynomial regression equation was obtained for estimating the supply airflow Q1e(ΔP, Vb) using the least-squares method:***Q*_1e_** (**Δ*P***, **V_b_**) = (−2.5926 × 10^−4^) **Δ*P***^2^ **V_b_**^2^ + (1.619 × 10^−2^) **Δ*P* V_b_**^2^ + (−1.3192 × 10^0^) **V_b_**^2^+ (7.6260 × 10^−3^) **Δ*P***^2^ **V_b_** + (−3.7841 × 10^−1^) **Δ*P* V_b_** + (5.1077 × 10^1^) **V_b_**+ (−5.7809 × 10^−2^) **Δ*P***^2^ + (1.0886 × 10^0^) **Δ*P*** + (−1.3114 × 10^2^)

The curves calculated using this regression equation at the three measured battery voltages (11.00, 12.00, and 13.00 V) are also shown in Figure 2.

For the exhaust filter (thin nonwoven fabric, one sheet attached to each of the left and right openings of the shield, W: 2.5 cm × H: 5.0 cm; total surface area ≈ 25 cm^2^), preliminary experiments confirmed that the exhaust flow rate ***Q*_2_** is proportional to the differential pressure **Δ*P***. However, significant time-dependent variation was observed. Therefore, in this prototype, the **Δ*P*** -dependence of the exhaust flow rate ***Q*_2_** was evaluated using the following procedure:(1)The prototype PAPR was fitted onto a mannequin head without gaps and activated.(2)Differential pressure **Δ*P*** and battery voltage **V_b_** were measured with *n* = 1000 data points at a sampling interval of approximately 15 ms. (This was simple data acquisition using the experimental system shown later in Figure 3, without employing the Respiratory Airflow Simulator.).(3)The estimated supply flow rate ***Q*_1e_**(**Δ*P***, **V_b_**) (defined as positive toward the inside of the hood) had a mean value of 241.29 L/min and a standard deviation of 2.97 L/min.(4)The measured differential pressure **Δ*P*** had a mean value of 28.21 Pa and a standard deviation of 1.37 Pa.(5)Assuming no leakage, the exhaust flow rate ***Q*_2e_**(**Δ*P***) (defined as positive outward from the hood) was taken to be equal to the supply flow rate ***Q*_1e_**(**Δ*P***, **V_b_**) (=241.29 L/min). From ***Q*_2e_**(**Δ*P***) = α**Δ*P***, the proportionality coefficient was obtained as α = ***Q*_2e_**(**Δ*P***)/**Δ*P*** = 241.29/28.21 = 8.55. Thus, ***Q*_2e_**(**Δ*P***) = 8.55**Δ*P*** [L/min].

This regression line is also shown in Figure 2. The intersection point between ***Q*_2e_**(**Δ*P***) and the ***Q*_1e_**(**Δ*P***, **V_b_**) curve represents the operating point of the PAPR under conditions where no other air exchange (e.g., breathing flow or leakage) occurs.

Both ***Q*_1e_**(**Δ*P***, **V_b_**) and ***Q*_2e_**(**Δ*P***) were derived under static conditions (i.e., constant fan output and steady flow). Assuming that there is no leakage from gaps between the hood seal and the user’s face, the following relationship holds among the estimated supply flow ***Q*_1e_**, exhaust flow ***Q*_2e_**, and respiratory (disturbance) flow ***Q*_3e_** (positive inward):***Q*_1e_**(**Δ*P***, **V_b_**) + ***Q*_3e_** = ***Q*_2e_**(**Δ*P***)***Q*_3e_** = ***Q*_2e_**(**Δ*P***) − ***Q*_1e_**(**Δ*P***, **V_b_**)

The operating range of the differential pressure **Δ*P*** inside the hood of the prototype PAPR was approximately 0 ≤ **Δ*P*** ≤ 50 Pa, which corresponds to less than 0.05% of atmospheric pressure (≈1.01 × 10^5^ Pa). Therefore, when compared with the measurement uncertainty of airflow in this experiment (approximately 1.5%), the influence of air compressibility can be considered sufficiently small, and it is appropriate to treat the flow as incompressible.

During respiratory activity, the estimated ***Q*_3e_** fluctuates periodically around zero. Its time-integrated value ∫***Q*_3e_** d***t*** also oscillates near zero, with the time average tending toward zero as the integration period increases.

However, when model errors exist in ***Q*_1e_** and ***Q*_2e_** or leakage flow occurs due to small gaps between the hood and face (as in this mannequin-based setup), this assumption no longer holds. In such cases, outward leakage from the hood occurs due to positive pressure, and this leakage is included in the computed ***Q*_3e_**.

Thus, when leakage is present, ***Q*_3e_** is biased in the negative direction, and the integral ∫***Q*_3e_** d**t** accumulates increasingly negative values over time.

In the controller program developed in this study, the wearing status (ON/OFF) and leak status (YES/NO) were determined according to the following procedure:

**(0) Setting of the judgment interval:** The controller continuously loops through calculations of various parameters and outputs them as datasets via serial communication. The judgment of wearing status and leak status is executed every 1000 datasets (corresponding to approximately 15 s in this experiment).

**(1) Judgment of wearing status:** In each judgment interval, if the maximum value of the differential pressure **Δ*P*** exceeds the threshold (10 Pa in this experiment), the wearing status is judged as ON (PAPR worn); otherwise, it is judged as OFF (not worn). Empirically, it was confirmed that even with small gaps during wearing, the maximum internal pressure consistently exceeded 10 Pa. In contrast, when the PAPR is not worn, the hood pressure is released to the atmosphere and remains nearly constant at ambient pressure, so **Δ*P*** never exceeds 10 Pa. Based on this empirical knowledge, the threshold was provisionally set to 10 Pa. However, this threshold was established for convenience, and future studies should set it based on more logical and systematic criteria.

**(2) Judgment of leak status:** Similarly, in each judgment interval, if the integrated value of respiratory flow (∫***Q*_3e_** d***t***) falls below the threshold (−5 L in this experiment), the leak status is judged as YES (leak present); otherwise, it is judged as NO (no leak). This threshold of −5 L was also determined empirically from preliminary experiments. Specifically, even when a straw was inserted into the sealing area of the hood to create a relatively small “leak,” values of at least −5 L were consistently observed, regardless of the start or end point of the integration interval. Therefore, the threshold was set to −5 L.

## 3. Evaluation of the Prototype PAPR System

### 3.1. Evaluation Apparatus and Method for the Prototype PAPR System

Figure 3 shows a schematic diagram of the Respiratory Airflow Testbench constructed for the functional evaluation of the prototype PAPR system. This testbench was designed to simulate the movement of air into and out of the PAPR hood caused by the wearer’s respiration. A mannequin head with an air supply tube penetrating through the mouth was fabricated. This tube was connected to a cylinder–piston drive operated manually by the experimenter, allowing the reproduction of airflow corresponding to breathing motion.

The airflow inside the tube was measured using a flow sensor (model: SFM3000-200-C, manufactured by Sensirion; measurement range: ±200 L/min, accuracy: 1.5%, response time: 0.5 ms, communication interface: I^2^C).

The dataset generated by the controller in each loop cycle consisted of the following parameters:

(1) Parameters measured, estimated, or determined by the PAPR controller.

(1.1) Measured values:

 Time: ***t*** [ms];

 Differential pressure: **Δ*P*** [Pa];

 Battery voltage: **V_b_** [V].

(1.2) Estimated values:

 Supply flow: ***Q*_1e_** (**Δ*P***, **V_b_**)’

 Exhaust flow: ***Q*_2e_** (**Δ*P***)’

 Disturbance flow: ***Q*_3e_** (calculated as ***Q*_3e_** = ***Q*_2e_** − ***Q*_1e_**).

(1.3) Determined Wearing Conditions:

 Wearing status (ON/OFF);

 Leak status (Yes/No).

(2) Parameters obtained from the flowmeter in the Respiratory Airflow Testbench:

Measured reference value;

 Disturbance flow: ***Q*_3m._**

This dataset was outputted from the controller via USB serial communication at approximately 15 ms intervals and transmitted to a PC for data logging.

The PC application used for data logging included input fields for the number of datasets before recording starts (Npre), the filename (Filename), and the number of datasets to be recorded (Nrecord), in addition to a “Start” button. The application also allowed setting the timing of two sound notifications (Nsound1, Nsound2) to alert the experimenter of condition changes.

The PAPR controller operated independently of the PC and was not affected by it. Therefore, the timing of recording start on the PC side corresponded to the point at which a specific dataset was received from the controller at the fixed 15 ms interval.

Figure 4 illustrates the full configuration of the prototype PAPR system connected with the Respiratory Airflow Testbench. The evaluation setup consists of the prototype PAPR, the Android smartphone running the “PAPR Wearing Status Monitor” application connected via BLE, and the externally connected testbench, which includes a cylinder–piston-type breathing simulator and a flow sensor.

During the functional evaluation experiments, the prototype PAPR system and its BLE-connected smartphone operated in their standard configuration. However, two additional tasks were implemented in the PAPR controller for experimental purposes:

[a] Acquisition of the measured disturbance flow ***Q*_3m_** from the flow sensor (I^2^C interface);

[b] Serial output via USB of operational parameters along with ***Q*_3m._**

As previously described, the prototype PAPR continuously estimated the respiratory flow ***Q*_3e_** and determined the wearing status (ON/OFF) and leak status (Yes/No) based on the differential pressure and battery voltage. These results, together with the measured ***Q*_3m_**, were sent to the PC via USB serial communication for logging and subsequent analysis.

The cylinder–piston drive was operated by manually moving an actuation rod back and forth. The experimenter adjusted the motion according to their own breathing rhythm to simulate realistic disturbance airflow. This airflow, entering the hood, was measured in real time by the flowmeter.

Meanwhile, the PAPR unit continued operating as usual, estimating ***Q*_3e_** and judging the wearing and leak status based on internal differential pressure and battery voltage. These results, together with the reference flow ***Q*_3m_**, were transmitted to the PC and used for evaluation and comparative analysis.

### 3.2. Performance Evaluation of the Prototype PAPR System

Figure 5 shows the time variation of the operating parameters of the prototype PAPR connected to the Respiratory Airflow Testbench. The respiratory flow (disturbance flow) ***Q*_3e_**, estimated solely from differential pressure **Δ*P*** [Pa] and battery voltage ***V*_b_** [V] by the PAPR controller, is shown by the red line. It demonstrates good agreement with the measured value ***Q*_3m_** obtained from the reference flow sensor (blue line).

The figure also displays the time variation of ***Q*_1e_** (estimated supply flow), ***Q*_2e_** (estimated exhaust flow), and **Δ*P***, which were used to calculate ***Q*_3e_**. These results confirm the effectiveness of the real-time respiratory flow estimation using a single differential-pressure sensor.

Figure 6 shows a scatter plot of the estimated respiratory flow ***Q*_3e_** (calculated solely from **Δ*P*** and **V_b_**) versus the measured value ***Q*_3m_** obtained from the flow sensor. The linear regression of the entire dataset yielded a slope of 1.154, an intercept of −4.809, and a coefficient of determination R^2^ = 0.952. In addition, supplementary evaluations of the relationship produced a correlation coefficient R = 0.976, a concordance correlation coefficient (CCC) = 0.960, a mean absolute error (MAE) = 15.94 L/min, and a root mean squared error (RMSE) = 19.00 L/min. These results indicate a high degree of agreement and reasonable estimation accuracy between ***Q*_3e_** and ***Q*_3m_**

Figure 7 illustrates the time variation of the time-integrated respiratory flow ***Q*_3e_**. The waveform of the integrated value ∫***Q*_3e_** d***t*** shows periodic fluctuations but trends downward over time. Assuming that ***Q*_3e_** is correctly estimated, this trend suggests the presence of a small leakage flow.

From the slope of the regression line (−0.077 L/s), the estimated leakage rate is approximately 4.6 L/min, which corresponds to about 1.9% of the average supply flow ***Q*_1e_** (240.0 L/min).

Although it is not possible to determine whether this leakage was due to actual physical leakage or modeling errors in ***Q*_1e_** and ***Q*_2e_**, the average and standard deviation of the differential pressure (μ = 27.6 Pa, σ = 7.29 Pa) indicate that a stable positive pressure was maintained inside the hood, and thus the PAPR functioned properly.

Figure 8 shows the time variation of operating parameters during an intentional manipulation of a gap. A large gap was created by inserting a straw into the sealing area. Data were recorded for 4000 samples (approximately 60 s). At sample number 2000 (approximately 29 s), a buzzer was sounded, and the experimenter removed the straw to close the gap.

From ***t*** = 0 to 30 s, the baseline of ***Q*_3e_** clearly shifts to the negative side, indicating the presence of continuous leakage flow. After ***t*** = 30 s, the baseline of ***Q*_3e_** returns to near zero, suggesting that the leakage was eliminated.

Even during the period when the gap was present, the differential pressure **Δ*P*** remained in the positive range, indicating that inward flow from outside the hood was prevented. These results imply that the positive-pressure design of the PAPR effectively prevents external air from entering the hood, even in the presence of some leakage, thereby maintaining its primary infection-prevention function.

Figure 9 illustrates the timing of automatic leak detection and smartphone display updates. The experiment started with a deliberately created large gap by inserting a straw into the seal area. Around ***t*** = 30 s, the straw was removed to close the gap.

As a result, the decreasing trend of the time-integrated ***Q*_3e_** (∫***Q*_3e_** d***t***) became more gradual, confirming the reduction in leakage. Leak detection was performed every judgment interval (1000 samples, approximately 15 s), and when ∫***Q*_3e_** d***t*** dropped below the threshold of −5 L, it was judged as “leak present.”

Therefore, the leak status update occurred around ***t*** = 40.7 s, and the smartphone display changed from blue (No Leak) to red (Leakage) at that point. Changes in the wearing status and leak status are shown by the green and red lines, respectively.

When leakage occurs, the integrated respiratory volume (∫***Q*_3e_** d***t***) decreases over time. In this study, the integrated respiratory volume was calculated for each evaluation interval (1000 samples, approximately 15 s), and if the value fell below the threshold of −5 L, the status was judged as “Leak present.” However, as shown in Figure 9, the integrated value strongly depends on the timing of the integration start and end points. For more accurate evaluation, a method that uses the slope of the median trend of the integrated respiratory volume would be preferable. Nevertheless, in this study, considering the computational capacity of the controller, the integrated value itself in each interval was used as the judgment parameter. In principle, leak detection should be realized by an algorithm that accurately captures the trend of the integrated respiratory volume—interpreting “no change” as ‘no leak’ and “decrease” as ‘leak present’.

In this study, leak detection was performed not by using the slope of a regression line but by the terminal value of the cumulative flow ***I*** = ∫***Q*_3e_** d***t*** (discrete integration) within each evaluation interval. Specifically, the integration value ***I*** was reset to zero at the beginning of each interval and updated cumulatively in every loop cycle. At the end of the interval, ***I*** was compared with the threshold of −5 L; if ***I*** fell below the threshold, the leak status was judged as YES, otherwise as NO (Evaluation was performed only during intervals where wearing status = ON).

Accordingly, this method corresponds to judging whether the integrated difference of ***Q*_3e_** over the interval exceeds 5 L, rather than evaluating the decrease amount over 15 s by a regression line. To reduce endpoint dependency, future improvements will consider a method that directly estimates the time trend (slope) of the integrated value.

Furthermore, in practical application environments, changes in the flow–pressure relationship model due to component failure or degradation of PAPR parts may cause false positives (judging as leak present when no leak exists) or false negatives (failing to detect an actual leak). Therefore, it will be an important future task to introduce on-site re-evaluation methods and dynamic correction algorithms for model characteristics. For example, mechanisms such as periodic zero-point calibration and self-diagnosis functions could be implemented to track model changes in real time, thereby reducing false alarms and maintaining the reliability of leak detection.

## 4. Discussion

### 4.1. Pressure Control to Assist Breathing Activity

This study experimentally confirmed that the estimated value of respiratory flow (disturbance flow) ***Q*_3e_**(**Δ*P***, **V_b_**) can be obtained with high accuracy using a prototype PAPR equipped with a differential-pressure sensor. This result forms a crucial foundation for the future implementation of a “pressure control function to assist breathing.”

Specifically, the following types of dynamic pressure control can be envisioned:(1)When an exhalation is detected, the differential pressure **Δ*P*** is decreased so as not to obstruct the exhalation.(2)When an inhalation is detected, **Δ*P*** is increased to assist inhalation.

However, several considerations must be taken into account when implementing such control:*Even during exhalation, a minimum level of positive pressure must be maintained so that the airflow direction always remains from inside to outside the hood, even if there is a small gap (possible leakage point). The minimum positive pressure required for this purpose needs to be experimentally determined in future work.*In pump control, there is a delay of approximately one second between a change in voltage and the corresponding change in actual flow rate. Therefore, if a command is issued only after detecting the respiratory phase, the control system may not be able to follow the breathing cycle adequately (typically about 5 s during rest, or about 4 s during light activity [21]).
Accordingly, two future control strategies are considered promising:
*The implementation of a feedforward control scheme based on predictive estimation of respiratory state.*The design of a “human–machine cooperative control” approach that assumes coordination between the wearer and the PAPR system.

The term “human–machine cooperative control” here refers to a control method in which the wearer becomes familiar with the behavior characteristics of the PAPR through a certain degree of training or acclimatization, allowing the device to respond more smoothly to the rhythm and timing of breathing. For example, if the PAPR can instantly adjust supply pressure in response to an intentionally modulated breathing pattern by the user, such “resonant control” may be one practical implementation of this concept.

To realize a practical pressure control function for breathing assistance, the control accuracy is expected to strongly depend on the following factors:(1)Control algorithm: Predictive control methods are required to compensate for the time delay between breath detection and control signal output, as well as the design of human–machine cooperative control schemes that assume coordination with the wearer.(2)Model accuracy: Unless the estimation model using the relationship between **Δ*P*** and respiratory flow ***Q*_3_** has sufficiently high accuracy (both static and dynamic), there is a risk of excessive or insufficient assistance.(3)Pump responsiveness: Since there is a delay of approximately 1 s from a voltage command to the corresponding change in airflow, the introduction of high-response pumps and/or algorithms to compensate for this delay is indispensable.

Addressing these technical issues will be an important future research task toward the practical implementation of pressure control for breathing assistance.

### 4.2. Calibration Method for Daily Use

In the daily use of PAPRs by the general public, regular maintenance is essential. However, under the “PAPR for Everyone” concept envisioned in this study, it is assumed that individuals wear PAPRs during outdoor activities and similar situations, thereby contributing to a reduction in airborne virus concentration in the environment. This could potentially eliminate the need for intensive disinfection and strict management as required in medical settings.

For instance, each PAPR constantly purifies air at a rate of approximately 200 L/min, which far exceeds the wearer’s resting respiratory flow (6–9 L/min). Most of the exhaust air is also discharged through an exhaust filter. Therefore, in environments where many people are wearing PAPRs, the overall concentration of airborne virus particles is expected to decrease naturally.

The key to the societal implementation of this technology is the development of a simple and field-applicable calibration method that can account for time-dependent changes in the fluid characteristics of the supply unit and exhaust filter, which are used in estimating the disturbance flow ***Q*_3_**. This is based on the estimation equation:***Q*_3e_**(**Δ*P***, **V_b_**) = ***Q*_2e_**(**Δ*P***) − ***Q*_1e_**(**Δ*P***, **V_b_**)

This section describes how to characterize and correct each component.

To calculate the estimated disturbance flow ***Q*_3e_**(**Δ*P***, **V_b_**), it is necessary to pre-characterize the supply flow characteristics of the intake unit, ***Q*_1e_**(**Δ*P***, **V_b_**), and the exhaust flow characteristics of the exhaust filter, ***Q*_2e_**(**Δ*P***). In this study, ***Q*_1e_**(**Δ*P***, **V_b_**) was first measured independently and its regression equation was derived.

Next, the PAPR was mounted on a mannequin head (standard model without penetration holes), and under the assumption of “no leakage,” ***Q*_2e_**(**Δ*P***) = ***Q*_1e_**(**Δ*P***, **V_b_**) was set. Based on this relationship, ***Q*_2e_**(**Δ*P***) was expressed in the form ***Q*_2e_**(**Δ*P***) = α·**Δ*P***, and the coefficient α was determined to be 8.55.

In practical operation, HEPA filters in the supply unit can be used continuously for several months. However, clogging or deformation due to external force may alter their characteristics. To accommodate such situations, the following procedure is proposed as a simple calibration method that can be conducted in a home environment:

(1) Self-diagnosis through open operation of the supply unit

The PAPR is operated in an open-air condition. If the HEPA filter in the supply unit is clogged, the pressure loss at the filter increases. Although the response may vary depending on the control method, under normal conditions the pump driving current ***I*_p_** changes in response to a given voltage. By pre-calibrating this relationship, it may be possible to estimate the degree of filter degradation (change in pressure loss characteristics) based on the voltage–current characteristics at the start of operation. Based on this, ***Q*_1e_**(**Δ*P***, **V_b_**) can theoretically be corrected. Ip can be easily measured by the controller (Arduino UNO R4 WiFi) as the terminal voltage across a standard resistor.

(2) Correction of ***Q*_2e_**(**Δ*P***) by fitting the PAPR on a mannequin or human subject

After correcting ***Q*_1e_**(**Δ*P***, **V_b_**), the PAPR is mounted on a mannequin head and ***Q*_2e_**(**Δ*P***) is redefined assuming “no leakage” by setting ***Q*_2e_**(**Δ*P***) = ***Q*_1e_**(**Δ*P***, **V_b_**). Alternatively, a human subject can wear the device and hold their breath for a short period to achieve the same result.

(3) Calibration warning based on operational monitoring

During regular use, continuous monitoring of operating parameters (**Δ*P***, **V_b_**, ***I*_p_**) in the supply unit can be used to detect changes in pressure loss characteristics. The system could then notify the user when steps (1) or (2) need to be performed.

Regarding the service life of HEPA filters, the U.S. Department of Energy (DOE) recommends the replacement of HEPA filters used for radioactive aerosol control in nuclear facilities every 10 years after manufacture. However, there have been reports of use beyond this duration [22]. Other sources suggest that the maximum safe usage period is 10 years under dry conditions and 5 years under humid conditions (e.g., spray or high humidity) [23]. In contrast, HEPA filters used in household air purifiers are generally recommended to be replaced every 6 months to 1 year.

In addition, the PAPR developed in this study has the potential to be applied to a mechanism for detecting time-dependent performance changes of HEPA filters. However, the detailed verification of this capability remains a future task. This is because the degradation factors of HEPA filter characteristics are not limited to simple clogging but may also include deformation due to external forces, the adhesion of organic substances, changes in electrostatic charge, and environmental humidity (the adsorption of water molecules), all of which act in combination. Consequently, the experimental conditions must be set with great care. At present, even the design of experiments comparing normal filters with intentionally clogged filters—aimed at evaluating detection capability through changes in related parameters—still poses technical challenges. Therefore, this paper does not address experimental demonstration of filter clogging detection, but instead identifies it as a future task, with plans to systematically classify degradation factors and design corresponding evaluation systems.

Furthermore, the long-term stability of the entire PAPR system must also be examined. In particular, during extended use, not only the clogging or deformation of the HEPA filter, as mentioned above, but also the degradation of pump components (such as the motor, bearings, and impeller), accumulation of dirt, and even time-dependent deterioration of electronic components (e.g., MOS-FETs for pump driving) may alter the fluid characteristics. Therefore, an important future task will be to conduct long-term stability tests that include not only the filter but also the electrical and mechanical components in order to evaluate the overall reliability of the system.

With regard to calibration methods for daily use, this paper remains at the conceptual level, and further verification is required concerning their feasibility and simplicity. In particular, from the perspective of user-friendliness, it remains unclear whether the proposed self-diagnosis and calibration methods are applicable to ordinary users and whether users can correctly maintain the equipment. Future work will involve verification through actual user trials and usability evaluations, with improvements made to the methods as necessary.

In addition, during long-term use or routine diagnostics, it is desirable to incorporate a mechanism that recognizes characteristic patterns of fluid property changes associated with typical errors—such as filter clogging, motor malfunction, and the damage or deformation of the flow path—and provides feedback to the user regarding the likely cause of the error. Such an error-feedback mechanism would encourage appropriate user responses and is essential for maintaining the reliability of the equipment.

### 4.3. Expansion to Biosensing and Environmental Sensing

Under the concept of “PAPR for Everyone,” the envisioned PAPR is a device equipped with power, computational capabilities, and a head-mounted support system, designed to be worn continuously by the public during outdoor activities in a pandemic situation. Such a continuously wearable PAPR has wide potential applications in both biosensing and environmental sensing.

As an example of biosensing, the differential-pressure sensor used in this study can be utilized for cough detection and respiratory pattern analysis, enabling the estimation of the user’s health status. In addition, various sensors that measure biometric information—such as body temperature, pulse, blood pressure, and brain activity—can also be integrated. Examples include temperature sensors, photoplethysmography (PPG) sensors, cuffless blood pressure estimation sensors, and EEG sensors. These could support daily health monitoring.

For environmental sensing, it is feasible to mount sensors that measure parameters such as the CO_2_ concentration, ambient temperature, humidity, illumination, particulate matter concentration (e.g., PM2.5), and noise level. In particular, CO_2_ concentration is a widely recognized indicator of indoor ventilation conditions and is strongly correlated with the volumetric concentration of exhaled aerosols and viral load in the air. As such, it can serve as a critical metric for evaluating infection risk [24,25].

Furthermore, in the field of gas detection technology, advances have been reported in high-sensitivity and high-selectivity sensors based on piezoelectric composites [26]. In the area of wearable biomonitoring, it has also been demonstrated that piezoresistive sensors based on nanofibers can sensitively detect respiratory patterns and pulse waves [27].

Furthermore, by constructing a network-based management system that collects and analyzes biosensing and environmental sensing data in real time from individual PAPR units, it may become possible to clarify infection mechanisms and establish more efficient control strategies through big data analytics. For instance, the system could function as a highly accurate contact tracing application. If abnormal respiratory patterns are detected in a specific area, the system could also assist in the early identification of infection clusters and the optimization of movement restrictions.

In such systems, where biometric and behavioral information is centrally collected, privacy protection and proper information governance are indispensable. In particular, when data are linked to personal identifiers or combined with location information, design and operational guidelines that ensure privacy—such as anonymization, decentralized management, and purpose limitation—will be required [28]. Furthermore, to handle networked data securely, standard cybersecurity measures such as authentication, encrypted communication, and intrusion detection must be combined. In applications such as real-time alerts or the visualization of infection risk, it is also crucial to keep the transmission and processing delays of sensor information within acceptable limits. These challenges remain important issues to be addressed for the societal implementation of the proposed system.

Differences in fit arising from variations in head size and shape are directly linked to leakage and can become a major source of error in the estimation of respiratory parameters. Therefore, the optimization of the hood, particularly the sealing section, is desirable. Specifically, by fabricating the sealing part with a 3D printer based on individual 3D shape measurements, it would be possible to realize a PAPR that ideally fits each user. If sufficient tightness is ensured and leakage is suppressed, the method proposed in this paper is expected to enable the robust and accurate estimation of respiratory parameters using only differential pressure and battery voltage. The data presented in this study represent typical measurement results, and no significant variations were observed across multiple trials, suggesting a certain degree of validity in terms of reproducibility. In future work, additional experiments will be conducted with representative user groups to statistically verify the reproducibility and reliability of the measurements, and based on the results, we plan to evaluate the model accuracy and develop correction methods as necessary.

The results presented in this paper represent typical measurement examples and were not arbitrarily selected. In the developed Respiratory Airflow Testbench (Figure 4), the cylinder–piston drive was operated manually to simulate respiratory airflow. However, with manual operation, it is difficult to systematically control respiratory amplitude and frequency. Therefore, future work will involve introducing a linear actuator drive to construct an experimental system capable of precisely controlling respiratory patterns. Furthermore, by equipping the system with two separate cylinder–piston drives for exhalation and inhalation; adjusting CO_2_ concentration, O_2_ concentration, and humidity on the exhalation side; and measuring the air composition on the inhalation side, it will be possible to reproduce (control and measure) the air circulation inside the hood. This approach is expected to enable the performance evaluation of the system under various workload and physiological conditions.

## 5. Conclusions

In this study, as part of a sustainable airborne infection control strategy to replace lockdowns, the author developed a prototype of a low-cost powered air-purifying respirator (PAPR) for the general public, equipped with real-time estimation functions for respiratory flow and mask leakage. This PAPR system accurately estimates respiratory (disturbance) flow ***Q*_3e_** using only a single differential-pressure sensor (SDP810) and battery voltage measurements and displays the wearing and leakage status on a smartphone via BLE communication.

In particular, Section 2 presented detailed measurements of the static flow characteristics of the supply and exhaust units and constructed regression models for estimating the supply flow ***Q*_1e_** and exhaust flow ***Q*_2e_** based solely on differential pressure and voltage. Using these models, ***Q*_3e_** was calculated, and an automatic judgment algorithm based on its time-integrated value was implemented, enabling leakage detection.

In Section 3, a new Respiratory Airflow Testbench with a cylinder–piston-driven breathing simulator was constructed. Under realistic wearing conditions, comparative evaluation between ***Q*_3e_** and the measured value ***Q*_3m_** obtained from a reference flow sensor was conducted. The results showed strong agreement, confirming the practicality of respiratory flow estimation and leakage detection using a single sensor. Furthermore, the time-integrated value ∫***Q*_3e_** d***t*** was shown to be useful for quantitatively estimating the magnitude of leakage flow.

Section 4 discussed future possibilities, including the implementation of dynamic pressure control synchronized with breathing activity, calibration methods for daily use, and the expansion to networked infection monitoring systems through the integration of biosensing and environmental sensing functions. Notably, it was suggested that personal PAPRs could serve as urban-scale infrastructure for visualizing infection risk by estimating health status from coughs and respiratory patterns and combining this with environmental data such as CO_2_ concentration and PM2.5.

This study demonstrated that the previously unresolved challenges of visualizing wearing conditions and detecting leakage in PAPRs can be achieved with a simple system configuration. The results provide a critical step toward balancing infection suppression with personal freedom of activity and offer a technological foundation for the societal deployment of citizen-oriented PAPRs in preparation for future pandemics.

## Figures and Tables

**Figure 1 sensors-25-05340-f001:**
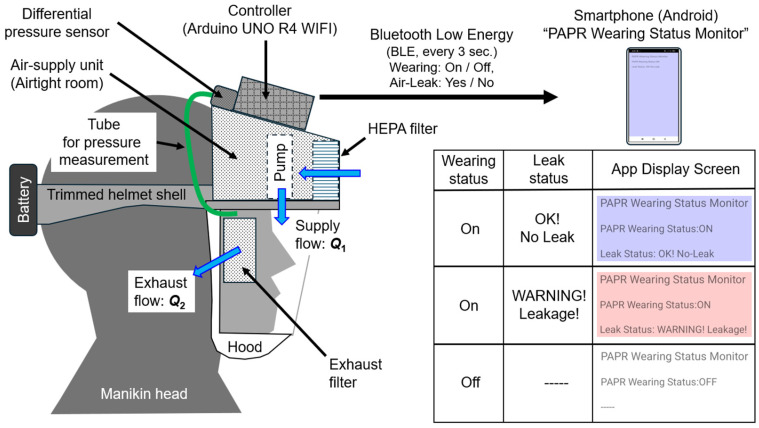
Prototype PAPR with respiratory flow estimation and leakage detection functions.

**Figure 2 sensors-25-05340-f002:**
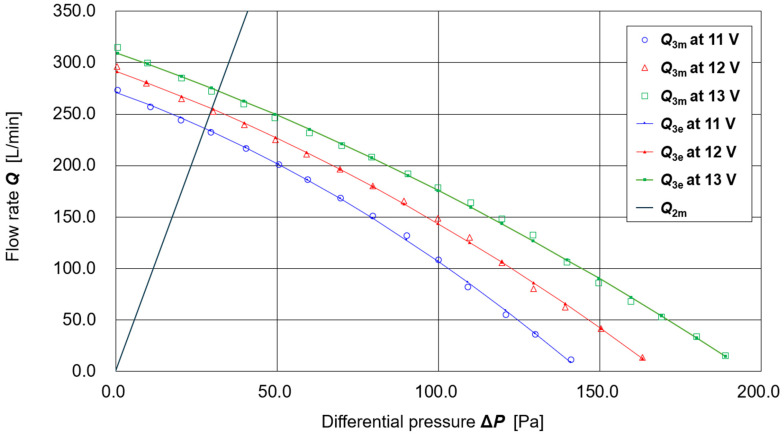
Flow characteristics of the supply unit and exhaust filter.

**Figure 3 sensors-25-05340-f003:**
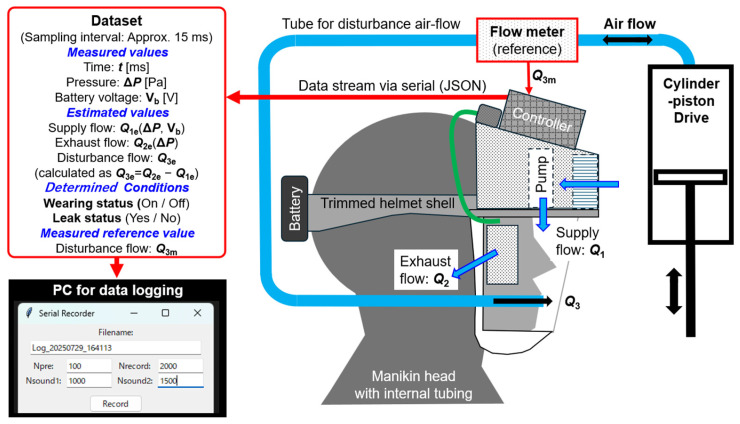
Respiratory Airflow Testbench for evaluating the functions of the prototype PAPR system. It consists of a mannequin head with a through-passing air supply tube, a flow sensor, and a cylinder–piston drive. It enables comparison between the estimated disturbance flow ***Q*_3e_** entering the PAPR hood and the measured value ***Q*_3m_**. The PAPR controller outputs datasets containing differential pressure **Δ*P***, battery voltage **V_b_**, estimated flows (***Q*_1e_**, ***Q*_2e_**, ***Q*_3e_**), measured flow ***Q*_3m_**, and wearing status (Wearing status, Leak status), which are logged on a PC.

**Figure 4 sensors-25-05340-f004:**
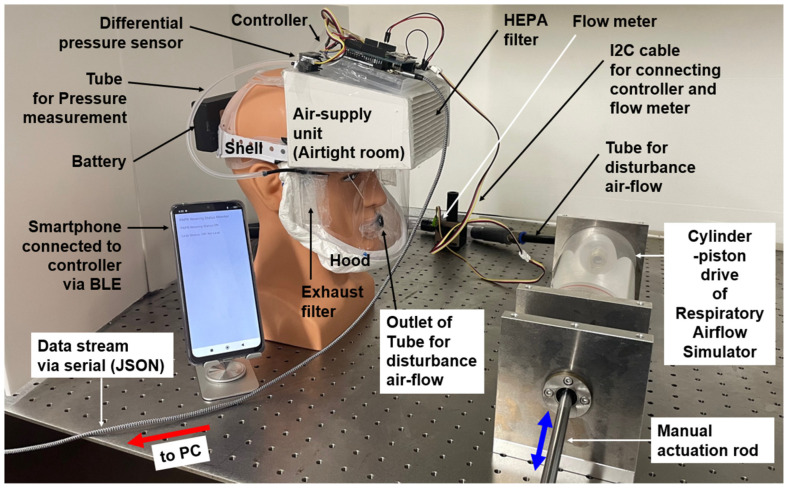
Overall configuration of the prototype PAPR system connected with the external Respiratory Airflow Testbench. The configuration includes the PAPR prototype itself, a BLE-connected smartphone, a breathing simulator, a flow sensor, and a PC.

**Figure 5 sensors-25-05340-f005:**
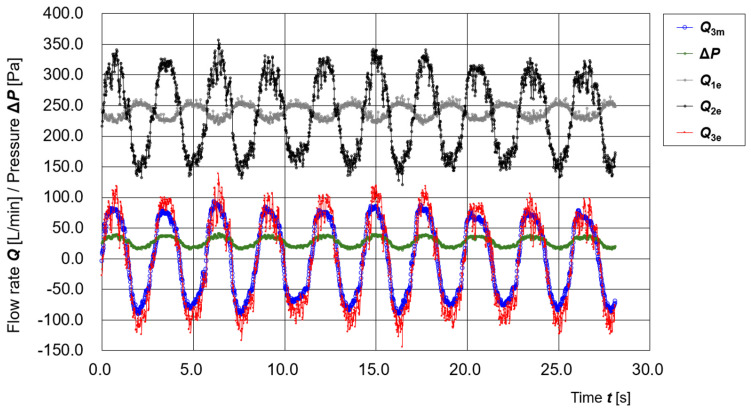
Time variation of operating parameters (***Q*_3m_**, **Δ*P***, ***Q*_1e_**, ***Q*_2e_**, ***Q*_3e_**) of the prototype PAPR connected to the Respiratory Airflow Testbench.

**Figure 6 sensors-25-05340-f006:**
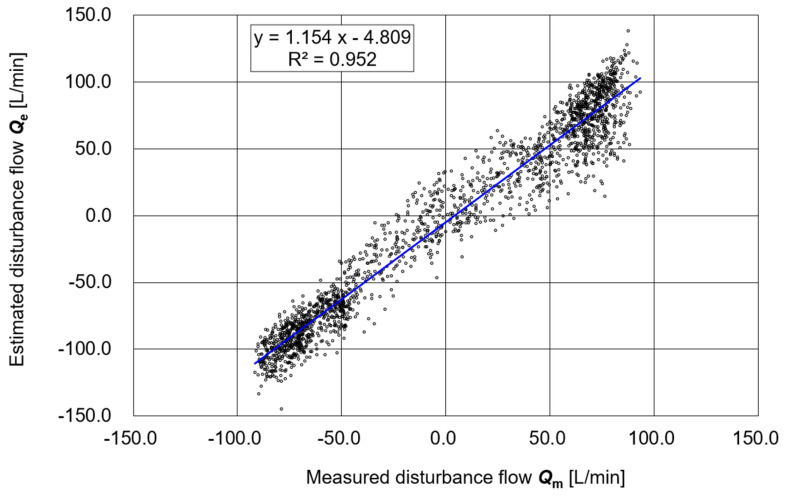
Correlation between the estimated value ***Q*_3e_** and the measured value ***Q*_3m_**.

**Figure 7 sensors-25-05340-f007:**
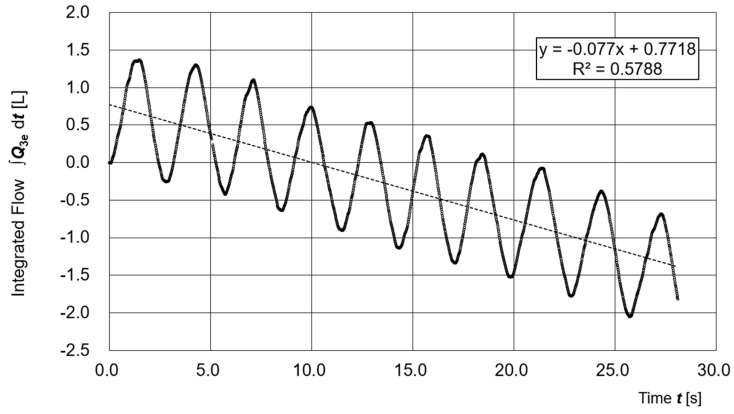
Time variation of the time-integrated respiratory flow ***Q*_3e_**.

**Figure 8 sensors-25-05340-f008:**
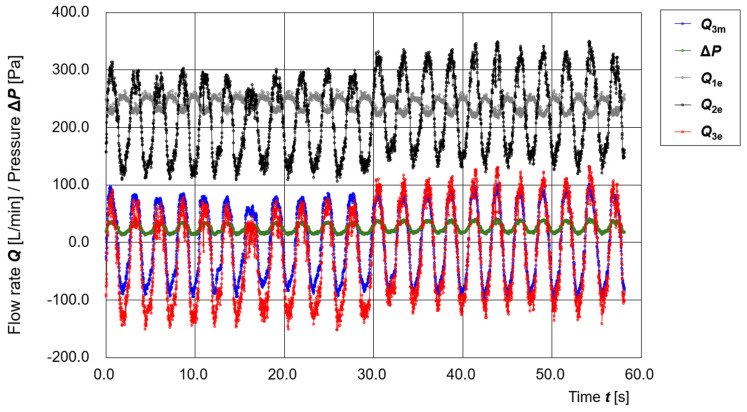
Time variation of operating parameters (***Q*_3m_**, **Δ*P***, ***Q*_1e_**, ***Q*_2e_**, ***Q*_3e_**) during intentional opening and closing of a gap.

**Figure 9 sensors-25-05340-f009:**
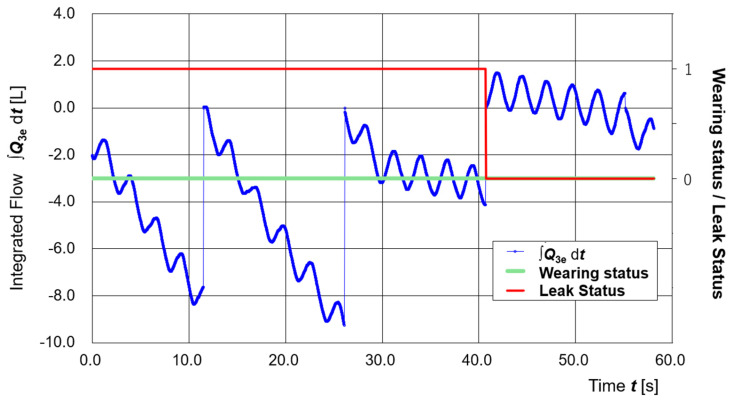
Time variation of the integrated respiratory volume (∫***Q*_3e_** d***t***: blue line), PAPR wearing status (green line), and leak status (red line).

## Data Availability

The datasets used and/or analyzed during the current study are available from the author on reasonable request.

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
