# Peer review of "The Real-Time Estimation of Respiratory Flow and Mask Leakage in a PAPR Using a Single Differential-Pressure Sensor and Microcontroller-Based Smartphone Interface in the Development of a Public-Oriented Powered Air-Purifying Respirator as an Alternative to Lockdown Measures"

_sensors, 2025, doi:10.3390/s25175340_

Round 1
Reviewer 1 Report
Comments and Suggestions for Authors
Please see attached.

Author Response
Response to Reviewer 1
I sincerely thank the reviewer for the constructive and detailed comments. Below I respond point by point and indicate the corresponding revisions in the manuscript.
1) On the derivation and calibration of α in Q₂e(ΔP)=αΔP
Comment: The coefficient α=8.55 is used without a clear calibration procedure or derivation.
Response: The following block has been revised as follows,
For the exhaust filter (thin nonwoven fabric, one sheet attached to each of the left and right openings of the shield, W: 2.5 cm × H: 5.0 cm; total surface area ≈ 25 cm²), preliminary experiments confirmed that the exhaust flow rate Q2 is proportional to the differential pressure ΔP. However, significant time-dependent variation was observed. Therefore, in this prototype, the ΔP -dependence of the exhaust flow rate Q2 was evaluated using the following procedure:
(1) The prototype PAPR was fitted onto a mannequin head without gaps and activated.
(2) Differential pressure ΔP and battery voltage Vb were measured with N = 1000 data points at a sampling interval of approximately 15 ms. (This was simple data acquisition using the experimental system shown later in Figure 3, without employing the Respiratory Airflow Simulator.)
(3) The estimated supply flow rate Q1e(ΔP, Vb) (defined as positive toward the inside of the hood) had a mean value of 241.29 L/min and a standard deviation of 2.97 L/min.
(4) The measured differential pressure ΔP had a mean value of 28.21 Pa and a standard deviation of 1.37 Pa.
(5) Assuming no leakage, the exhaust flow rate Q2e(ΔP) (defined as positive outward from the hood) was taken to be equal to the supply flow rate Q1e(ΔP, Vb) (= 241.29 L/min). From Q2e(ΔP) = αΔP, the proportionality coefficient was obtained as α = Q2e(ΔP)/ ΔP = 241.29/28.21 = 8.55. Thus, Q2e(ΔP) = 8.55ΔP [L/min].
2) Figure mis-reference (“Figure 4” vs “Figure 2”)
Comment: “This regression line is also shown in Figure 4” appears to be a writing error.
Response: Corrected to “Figure 2” in the revised text to match the actual figure content.
3) Grammar and typographical issues
Comment: Examples include “Q3em” (should be Q3m), and a missing verb in the Data Availability sentence.
Response: I have corrected the errors.
4) Justification for the 10 Pa threshold (Wearing status)
Comment: The explanation is vague; please justify.
Response: The following block has been revised in Chapter-2.
In the controller program developed in this study, the wearing status (ON / OFF) and leak status (YES / NO) were determined according to the following procedure:
[0] Setting of the judgment interval: The controller continuously loops through calculations of various parameters and outputs them as datasets via serial communication. The judgment of wearing status and leak status is executed every 1,000 datasets (corresponding to approximately 15 s in this experiment).
[1] Judgment of wearing status: In each judgment interval, if the maximum value of the differential pressure ΔP exceeds the threshold (10 Pa in this experiment), the wearing status is judged as ON (PAPR worn); otherwise, it is judged as OFF (not worn). Empirically, it was confirmed that even with small gaps during wearing, the maximum internal pressure consistently exceeded 10 Pa. In contrast, when the PAPR is not worn, the hood pressure is released to the atmosphere and remains nearly constant at ambient pressure, so ΔP never exceeds 10 Pa. Based on this empirical knowledge, the threshold was provisionally set to 10 Pa. However, this threshold was established for convenience, and future studies should set it based on more logical and systematic criteria.
[2] Judgment of leak status: Similarly, in each judgment interval, if the integrated value of respiratory flow (∫Q₃e dt) falls below the threshold (−5 L in this experiment), the leak status is judged as YES (leak present); otherwise, it is judged as NO (no leak). This threshold of −5 L was also determined empirically from preliminary experiments. Specifically, even when a straw was inserted into the sealing area of the hood to create a relatively small “leak,” values of at least −5 L were consistently observed, regardless of the start or end point of the integration interval. Therefore, the threshold was set to −5 L.
5) Rationale for the −5 L leakage threshold
Comment: Why −5 L?
Response-1: The following block has been revised in Chapter-2.
In the controller program developed in this study, the wearing status (ON / OFF) and leak status (YES / NO) were determined according to the following procedure:
[0] Setting of the judgment interval: The controller continuously loops through calculations of various parameters and outputs them as datasets via serial communication. The judgment of wearing status and leak status is executed every 1,000 datasets (corresponding to approximately 15 s in this experiment).
[1] Judgment of wearing status: In each judgment interval, if the maximum value of the differential pressure ΔP exceeds the threshold (10 Pa in this experiment), the wearing status is judged as ON (PAPR worn); otherwise, it is judged as OFF (not worn). Empirically, it was confirmed that even with small gaps during wearing, the maximum internal pressure consistently exceeded 10 Pa. In contrast, when the PAPR is not worn, the hood pressure is released to the atmosphere and remains nearly constant at ambient pressure, so ΔP never exceeds 10 Pa. Based on this empirical knowledge, the threshold was provisionally set to 10 Pa. However, this threshold was established for convenience, and future studies should set it based on more logical and systematic criteria.
[2] Judgment of leak status: Similarly, in each judgment interval, if the integrated value of respiratory flow (∫Q₃e dt) falls below the threshold (−5 L in this experiment), the leak status is judged as YES (leak present); otherwise, it is judged as NO (no leak). This threshold of −5 L was also determined empirically from preliminary experiments. Specifically, even when a straw was inserted into the sealing area of the hood to create a relatively small “leak,” values of at least −5 L were consistently observed, regardless of the start or end point of the integration interval. Therefore, the threshold was set to −5 L.
Response-2: The following block has been added in the end of Chapter-3.
When leakage occurs, the integrated respiratory volume (∫Q3e dt) decreases over time. In this study, the integrated respiratory volume was calculated for each evaluation interval (1,000 samples, approximately 15 s), and if the value fell below the threshold of −5 L, the status was judged as “Leak present.” However, as shown in Figure 9, the integrated value strongly depends on the timing of the integration start and end points. For more accurate evaluation, a method that uses the slope of the median trend of the integrated respiratory volume would be preferable. Nevertheless, in this study, considering the computational capacity of the controller, the integrated value itself in each interval was used as the judgment parameter. In principle, leak detection should be realized by an algorithm that accurately captures the trend of the integrated respiratory volume—interpreting “no change” as no leak and “decrease” as leak present.
6) Incompressible-flow assumption
Comment: Please verify the validity of treating the flow as incompressible.
Response: I added a theoretical justification with quantitative comparison.
The operating range of the differential pressureΔP inside the hood of the prototype PAPR was approximately 0 ≤ ΔP ≤ 50 Pa, which corresponds to less than 0.05% of atmospheric pressure (≈1.01 × 10⁵ Pa). Therefore, when compared with the measurement uncertainty of airflow in this experiment (approximately 1.5%), the influence of air compressibility can be considered sufficiently small, and it is appropriate to treat the flow as incompressible.
7) Add supplementary accuracy metrics (CCC, MAE, RMSE)
Comment: Consider CCC, MAE, RMSE in addition to R².
Response: I computed and reported R, CCC, MAE, RMSE for Figure 6.
Figure 6 shows a scatter plot of the estimated respiratory flow Q3e (calculated solely from ΔP and Vb) versus the measured value Q3m obtained from the flow sensor. Linear regression of the entire dataset yielded a slope of 1.154, an intercept of −4.809, and a coefficient of determination R² = 0.952. In addition, supplementary evaluations of the relationship produced a correlation coefficient R = 0.976, a concordance correlation coefficient (CCC) = 0.960, a mean absolute error (MAE) = 15.94 L/min, and a root mean squared error (RMSE) = 19.00 L/min. These results indicate a high degree of agreement and reasonable estimation accuracy between Q3e and Q3m.
8) Clarify the leakage judgment calculation (slope vs integral)
Comment: Is the −5 L criterion based on a regression-line drop, or the integral difference over ~15 s, or something else?
Response: I clarified the algorithm and emphasized it is not slope-based in Chapter-3.
In this study, leak detection was performed not by using the slope of a regression line but by the terminal value of the cumulative flow I=∫Q3e dt (discrete integration) within each evaluation interval. Specifically, the integration value I was reset to zero at the beginning of each interval and updated cumulatively in every loop cycle. At the end of the interval, I was compared with the threshold of −5 L; if I fell below the threshold, the leak status was judged as YES, otherwise as NO. (Evaluation was performed only during intervals where Wearing status = ON.)
Accordingly, this method corresponds to judging whether the integrated difference of Q3e over the interval exceeds 5 L, rather than evaluating the decrease amount over 15 s by a regression line. To reduce endpoint dependency, future improvements will consider a method that directly estimates the time trend (slope) of the integrated value.
9) Experimental verification of HEPA filter clogging detection
Comment: Please add experiments comparing normal vs clogged HEPA filters.
Response: I softened the claim and positioned it as future work, explaining why a rigorous testbed is non-trivial. The following block has been added at the end of Section 4.2.
In addition, the PAPR developed in this study has the potential to be applied to a mechanism for detecting time-dependent performance changes of HEPA filters. However, detailed verification of this capability remains a future task. This is because the degradation factors of HEPA filter characteristics are not limited to simple clogging, but may also include deformation due to external forces, adhesion of organic substances, changes in electrostatic charge, and environmental humidity (adsorption of water molecules), all of which act in combination. Consequently, the experimental conditions must be set with great care. At present, even the design of experiments comparing normal filters with intentionally clogged filters—aimed at evaluating detection capability through changes in related parameters—still poses technical challenges. Therefore, this paper does not address experimental demonstration of filter clogging detection, but instead identifies it as a future task, with plans to systematically classify degradation factors and design corresponding evaluation systems.
10) Long-term stability and drift
Comment: Add long-term stability tests considering battery aging, HEPA clogging, etc.
Response: The following block has been added at the end of Section 4.2.
Furthermore, the long-term stability of the entire PAPR system must also be examined. In particular, during extended use, not only clogging or deformation of the HEPA filter as mentioned above, but also degradation of pump components (such as the motor, bearings, and impeller), accumulation of dirt, and even time-dependent deterioration of electronic components (e.g., MOS-FETs for pump driving) may alter the fluid characteristics. Therefore, an important future task will be to conduct long-term stability tests that include not only the filter but also the electrical and mechanical components, in order to evaluate the overall reliability of the system.
I sincerely appreciate the reviewer’s insightful suggestions, which have greatly improved the clarity and rigor of the manuscript.

Reviewer 2 Report
Comments and Suggestions for Authors
The author presents a study involving the application of a single differential pressure sensor within a powered air purifying respirator (PAPR) system to enable real-time estimation of respiratory flow and mask leakage. The manuscript is well-structured and supported by extensive experimental validations. However, several concerns remain that should be addressed to improve the clarity and practical significance of the work:
- It is unclear whether the sensors were integrated into a commercial respirator or a fully customized system. Additionally, the manuscript lacks quantitative information regarding the weight added by the sensor system. How does the modified device compare to standard PPE in terms of overall weight? Is it comfortable and practical for extended use (e.g., 6-hour wear duration)? Although the authors note that a work helmet was partially cut and modified, specific weight measurements before and after modification are missing and should be provided.
- The manuscript would benefit from a clearer discussion of the practical significance of this system in clinical or industrial settings. Specifically, how does this solution compare to existing protective equipment, such as positive pressure hoods or N95 respirators, in terms of performance, cost, usability, or safety? Including comparisons with related works in the Introduction would contextualize the novelty and relevance of the proposed approach.
- Can the system robustly and accurately detect respiratory parameters across users with different facial geometries (e.g., varying face sizes and shapes)? Additional experiments or supporting data are needed to demonstrate the repeatability and reliability of the measurements across a representative user population.
- The current dataset used for validating breathing detection appears limited. To strengthen the claim that the sensor system can accurately detect breathing under realistic conditions, additional tests incorporating a range of breathing amplitudes and frequencies should be included. This will help assess the system’s performance under various workload and physiological conditions.
Author Response
Response to Reviewer 2
I sincerely thank the reviewer for the constructive and detailed comments. Below I respond point by point and indicate the corresponding revisions in the manuscript.
1) On whether the system is commercial or customized, and the total weight of the prototype
Comment: It is unclear whether the sensors were integrated into a commercial respirator or a customized system. Quantitative information about the added weight is missing. How does it compare to standard PPE? Is it comfortable for extended use (e.g., 6 hours)?
Response: The following block has been added in the middle of Chapter-2.
The prototype PAPR was developed by the author as a custom system assembled from commercially available components. Its total mass was approximately 656.1 g, including the battery (174.4 g), and the continuous operating time was about 320 minutes (5 h 20 min). The operating time can be extended depending on the battery capacity, and even for this prototype, more than 6 h of use is expected with capacity expansion.
For comparison, the commercially available PAPR “Versaflo TR-300+” manufactured by 3M (listed in [19], Table 1 [A]) has a total mass of approximately 1419.6 g, consisting of the belt-mounted air-supply unit (about 1107.7 g), the supply tube (about 201.9 g), and the hood (about 110.0 g). The fact that the mass of the prototype PAPR is less than half of this value is attributed to its integrated helmet-type structure and the use of lightweight materials, which contribute to improved comfort during long-term wear. However, for product development, it will be necessary to ensure an optimal balance between strength and weight.
2) Practical significance and comparison with existing protective equipment
Comment: The manuscript needs a clearer discussion of practical significance in clinical or industrial settings, and comparisons with existing PPE (positive pressure hoods, N95 respirators).
Response: The following block has been added in the second half of Chapter-1.
Although PAPRs provide high protective performance, it is important to clarify their societal significance by comparing them with other common protective equipment. The prototype PAPR developed in this study has unique value not only in terms of performance but also in cost and usability compared with existing N95 masks and medical-grade PAPRs. N95 masks are inexpensive and widely used, but they face challenges in terms of fit and comfort during long-term wear. On the other hand, commercial medical-grade PAPRs provide high protection but are priced on the order of 1,000 USD, and their weight and wearing configuration impose limitations on daily use by the general public.
The prototype PAPR presented in this study maintains protective performance equivalent to that of medical-grade PAPRs while ensuring comfort through weight reduction and structural simplification, and achieving low cost by utilizing commercially available components. In addition, the positive-pressure design reduces breathing resistance, thereby balancing user safety with reduced physiological burden. As a result, this system opens the possibility of a sustainable protective device suitable not only for medical settings but also for industrial environments and daily use by the general public.
3) Applicability across users with different facial geometries
Comment: Can the system detect respiratory parameters robustly across users with different face shapes and sizes? Additional supporting data are needed.
Response: I acknowledged this limitation and added a discussion in Section 4.3
Differences in fit arising from variations in head size and shape are directly linked to leakage and can become a major source of error in the estimation of respiratory parameters. Therefore, optimization of the hood, particularly the sealing section, is desirable. Specifically, by fabricating the sealing part with a 3D printer based on individual 3D shape measurements, it would be possible to realize a PAPR that ideally fits each user. If sufficient tightness is ensured and leakage is suppressed, the method proposed in this paper is expected to enable robust and accurate estimation of respiratory parameters using only differential pressure and battery voltage. The data presented in this study represent typical measurement results, and no significant variations were observed across multiple trials, suggesting a certain degree of validity in terms of reproducibility. In future work, additional experiments will be conducted with representative user groups to statistically verify the reproducibility and reliability of the measurements, and based on the results, we plan to evaluate the model accuracy and develop correction methods as necessary.
4) Limited dataset for breathing detection
Comment: The dataset used for validation appears limited. Additional tests with a range of breathing amplitudes and frequencies are needed to strengthen the claims.
Response: The following block has been added in Section 4.3.
The results presented in this paper represent typical measurement examples and were not arbitrarily selected. In the developed Respiratory Airflow Testbench (Figure 4), the cylinder-piston drive was operated manually to simulate respiratory airflow. However, with manual operation it is difficult to systematically control respiratory amplitude and frequency. Therefore, future work will involve introducing a linear actuator drive to construct an experimental system capable of precisely controlling respiratory patterns. Furthermore, by equipping the system with two separate cylinder-piston drives for exhalation and inhalation, adjusting CO₂ concentration, O₂ concentration, and humidity on the exhalation side, and measuring the air composition on the inhalation side, it will be possible to reproduce (control and measure) the air circulation inside the hood. This approach is expected to enable performance evaluation of the system under various workload and physiological conditions.
I sincerely appreciate the reviewer’s insightful suggestions, which have significantly improved the clarity and practical relevance of the manuscript.

Reviewer 3 Report
Comments and Suggestions for Authors
This paper presents a prototype of a low-cost powered air-purifying respirator (PAPR), with the aim of providing the public with an alternative option to lockdown measures. Nevertheless, several issues still remain to be resolved and refined.
- The paper mentions the detection of leakage through time integration Q3e, but whether the threshold setting of this method (such as -5 L) is reasonable has not been explained in detail. For instance, in the case of false alarms, in practical applications, environmental factors (such as changes in air flow) may lead to false positive or false negative results. The paper does not provide countermeasures or further verification.
- The paper proposes that pressure control functions may be realized in the future to assist breathing, but it does not elaborate on its technical feasibility in detail.
- The paper discusses the calibration methods in daily use, but their feasibility and simplicity need further verification. In terms of user-friendliness, it is not yet clear whether the suggested self-diagnosis and calibration methods are applicable to ordinary users, which may lead to difficulties for users to maintain the equipment correctly.
- Section 4.3 suggests networking respiratory characteristics and environmental data. However, data encryption, privacy protection and latency tolerance were not discussed.
- The paper proposes a simple "open-circuit self-check" and "mask airtight" calibration (P. 14-15), but lacks an error feedback mechanism.
- This work investigated the wearable sensors for respiratory monitoring. Some relative papers may enrich the concepts and background of this work as references: J. Mater. Chem. C, 2025,13, 13582-13606; J. Compos. Sci. 2025, 9(7), 339.
Author Response
Response to Reviewer 3
I sincerely thank the reviewer for the constructive and detailed comments. Below I respond point by point and indicate the corresponding revisions in the manuscript.
1) Threshold setting for leakage detection (−5 L) and risk of false positives/negatives
Comment: The threshold of −5 L for leakage detection is not well explained. Environmental factors may cause false alarms, and no countermeasures are discussed.
Response-1: The following blocks have been revised/added at the end of Chapter-2.
In the controller program developed in this study, the wearing status (ON / OFF) and leak status (YES / NO) were determined according to the following procedure:
[0] Setting of the judgment interval: The controller continuously loops through calculations of various parameters and outputs them as datasets via serial communication. The judgment of wearing status and leak status is executed every 1,000 datasets (corresponding to approximately 15 s in this experiment).
[1] Judgment of wearing status: In each judgment interval, if the maximum value of the differential pressure ΔP exceeds the threshold (10 Pa in this experiment), the wearing status is judged as ON (PAPR worn); otherwise, it is judged as OFF (not worn). Empirically, it was confirmed that even with small gaps during wearing, the maximum internal pressure consistently exceeded 10 Pa. In contrast, when the PAPR is not worn, the hood pressure is released to the atmosphere and remains nearly constant at ambient pressure, so ΔP never exceeds 10 Pa. Based on this empirical knowledge, the threshold was provisionally set to 10 Pa. However, this threshold was established for convenience, and future studies should set it based on more logical and systematic criteria.
[2] Judgment of leak status: Similarly, in each judgment interval, if the integrated value of respiratory flow (∫Q₃e dt) falls below the threshold (−5 L in this experiment), the leak status is judged as YES (leak present); otherwise, it is judged as NO (no leak). This threshold of −5 L was also determined empirically from preliminary experiments. Specifically, even when a straw was inserted into the sealing area of the hood to create a relatively small “leak,” values of at least −5 L were consistently observed, regardless of the start or end point of the integration interval. Therefore, the threshold was set to −5 L.
Response-2: The following blocks have been added at the end of Chapter-3.
When leakage occurs, the integrated respiratory volume (∫Q3e dt) decreases over time. In this study, the integrated respiratory volume was calculated for each evaluation interval (1,000 samples, approximately 15 s), and if the value fell below the threshold of −5 L, the status was judged as “Leak present.” However, as shown in Figure 9, the integrated value strongly depends on the timing of the integration start and end points. For more accurate evaluation, a method that uses the slope of the median trend of the integrated respiratory volume would be preferable. Nevertheless, in this study, considering the computational capacity of the controller, the integrated value itself in each interval was used as the judgment parameter. In principle, leak detection should be realized by an algorithm that accurately captures the trend of the integrated respiratory volume—interpreting “no change” as no leak and “decrease” as leak present.
Furthermore, in practical application environments, changes in the flow–pressure relationship model due to component failure or degradation of PAPR parts may cause false positives (judging as leak present when no leak exists) or false negatives (failing to detect an actual leak). Therefore, it will be an important future task to introduce on-site re-evaluation methods and dynamic correction algorithms for model characteristics. For example, mechanisms such as periodic zero-point calibration and self-diagnosis functions could be implemented to track model changes in real time, thereby reducing false alarms and maintaining the reliability of leak detection.
2) Feasibility of pressure control for breathing assistance
Comment: The feasibility of implementing pressure control functions is not elaborated.
Response: The following block has been added at the end of Section 4.1.
To realize a practical pressure control function for breathing assistance, the control accuracy is expected to strongly depend on the following factors:
(1) Control algorithm: Predictive control methods are required to compensate for the time delay between breath detection and control signal output, as well as the design of human–machine cooperative control schemes that assume coordination with the wearer.
(2) Model accuracy: Unless the estimation model using the relationship between ΔP and respiratory flow Q3 has sufficiently high accuracy (both static and dynamic), there is a risk of excessive or insufficient assistance.
(3) Pump responsiveness: Since there is a delay of approximately 1 s from a voltage command to the corresponding change in airflow, the introduction of high-response pumps and/or algorithms to compensate for this delay is indispensable.
Addressing these technical issues will be an important future research task toward the practical implementation of pressure control for breathing assistance.
3) Calibration methods for daily use and user-friendliness
Comment: The feasibility and simplicity of the proposed calibration methods require further verification.
Response: The following block has been added at the end of Section 4.2.
With regard to calibration methods for daily use, this paper remains at the conceptual level, and further verification is required concerning their feasibility and simplicity. In particular, from the perspective of user-friendliness, it remains unclear whether the proposed self-diagnosis and calibration methods are applicable to ordinary users and whether users can correctly maintain the equipment. Future work will involve verification through actual user trials and usability evaluations, with improvements made to the methods as necessary.
4) Networking of respiratory and environmental data (privacy and latency)
Comment: Section 4.3 does not address data encryption, privacy protection, or latency.
Response: The following block has been revised at the end of Section 4.3.
In such systems, where biometric and behavioral information is centrally collected, privacy protection and proper information governance are indispensable. In particular, when data are linked to personal identifiers or combined with location information, design and operational guidelines that ensure privacy—such as anonymization, decentralized management, and purpose limitation—will be required [28]. Furthermore, to handle networked data securely, standard cybersecurity measures such as authentication, encrypted communication, and intrusion detection must be combined. In applications such as real-time alerts or visualization of infection risk, it is also crucial to keep the transmission and processing delays of sensor information within acceptable limits. These challenges remain important issues to be addressed for the societal implementation of the proposed system.
5) Error-feedback mechanism for calibration
Comment: The paper lacks an error-feedback mechanism in the proposed calibration process.
Response: The following block has been added at the end of Section 4.2.
In addition, the PAPR developed in this study has the potential to be applied to a mechanism for detecting time-dependent performance changes of HEPA filters. However, detailed verification of this capability remains a future task. This is because the degradation factors of HEPA filter characteristics are not limited to simple clogging, but may also include deformation due to external forces, adhesion of organic substances, changes in electrostatic charge, and environmental humidity (adsorption of water molecules), all of which act in combination. Consequently, the experimental conditions must be set with great care. At present, even the design of experiments comparing normal filters with intentionally clogged filters—aimed at evaluating detection capability through changes in related parameters—still poses technical challenges. Therefore, this paper does not address experimental demonstration of filter clogging detection, but instead identifies it as a future task, with plans to systematically classify degradation factors and design corresponding evaluation systems.
Furthermore, the long-term stability of the entire PAPR system must also be examined. In particular, during extended use, not only clogging or deformation of the HEPA filter as mentioned above, but also degradation of pump components (such as the motor, bearings, and impeller), accumulation of dirt, and even time-dependent deterioration of electronic components (e.g., MOS-FETs for pump driving) may alter the fluid characteristics. Therefore, an important future task will be to conduct long-term stability tests that include not only the filter but also the electrical and mechanical components, in order to evaluate the overall reliability of the system.
With regard to calibration methods for daily use, this paper remains at the conceptual level, and further verification is required concerning their feasibility and simplicity. In particular, from the perspective of user-friendliness, it remains unclear whether the proposed self-diagnosis and calibration methods are applicable to ordinary users and whether users can correctly maintain the equipment. Future work will involve verification through actual user trials and usability evaluations, with improvements made to the methods as necessary.
In addition, during long-term use or routine diagnostics, it is desirable to incorporate a mechanism that recognizes characteristic patterns of fluid property changes associated with typical errors—such as filter clogging, motor malfunction, and damage or deformation of the flow path—and provides feedback to the user regarding the likely cause of the error. Such an error-feedback mechanism would encourage appropriate user responses and is essential for maintaining the reliability of the equipment.
6) Additional references on wearable sensors for respiratory monitoring
Comment: Relevant recent works should be cited.
Response: The following block has been added in the middle of Section 4.3.
Furthermore, in the field of gas detection technology, advances have been reported in high-sensitivity and high-selectivity sensors based on piezoelectric composites [26]. In the area of wearable biomonitoring, it has also been demonstrated that piezoresistive sensors based on nanofibers can sensitively detect respiratory patterns and pulse waves [27].
[26] W. Li, G. Xie, X. Huo, L. Que, H. Tai, Y. Jiang, and Y. Su, “Piezoelectric composites for gas sensing: evolution of sensing and transduction designs,” J. Mater. Chem. C, vol. 13, pp. 13582–13606, 2025.
https://doi.org/10.1039/D5TC01383F
[27] H. Pan, Y. Wang, G. Xie, C. Chen, H. Li, F. Wu, and Y. Su, “Flexible piezoresistive sensors based on PANI/rGO@PDA/PVDF nanofiber for wearable biomonitoring,” J. Compos. Sci., vol. 9, no. 7, p. 339, 2025.
https://doi.org/10.3390/jcs9070339
I sincerely appreciate the reviewer’s insightful suggestions, which have significantly improved the clarity and practical relevance of the manuscript.

Round 2
Reviewer 1 Report
Comments and Suggestions for Authors
All comments have been adequately addressed.
Reviewer 2 Report
Comments and Suggestions for Authors
The author has now addressed all my concerns.